# An Efficient and Simple Method for Collecting Haemolymph of Cerambycidae (Insecta: Coleoptera) Adults

**DOI:** 10.3390/insects14010029

**Published:** 2022-12-27

**Authors:** Yiming Niu, Yuxuan Zhao, Fengming Shi, Meng Li, Sainan Zhang, Jinglin Yang, Shixiang Zong, Jing Tao

**Affiliations:** 1Beijing Key Laboratory for Forest Pest Control, Beijing Forestry University, Beijing 100083, China; 2Mentougou Forestry Station, Beijing 102308, China

**Keywords:** cerambycid beetle, Cerambycidae, forestry pest, haemolymph, centrifugation

## Abstract

**Simple Summary:**

The lack of efficient methods to extract haemolymph from adult cerambycid beetles limits physiological and biochemical research on these important pest species. To enable the collection of large amounts of pure haemolymph from Cerambycidae adults efficiently and easily, we developed an innovative extraction method, termed net centrifugation. *Anoplophora chinensis*, *Monochamus saltuarius* and *Saperda populnea*, three cerambycid beetle species with substantial differences in adult size, were used to evaluate the newly developed method. Compared with estimates for other commonly used methods for coleopteran haemolymph collection, this method generated a larger amount of pure haemolymph in a shorter time. The net centrifugation method is therefore highly efficient, providing a solid foundation for further studies of cerambycid beetles.

**Abstract:**

Cerambycid beetles (Cerambycidae) are major forest pests, posing a serious threat to the security of forest resources worldwide. Extensive research has focused on the control of cerambycid beetles from physiological and biochemical perspectives. Despite the important roles of insect haemolymph in physiological processes, efficient collection methods for Cerambycidae are lacking. For the efficient and easy collection of large amounts of pure haemolymph from adult cerambycid beetles, a new method, named net centrifugation, was developed. Three species of cerambycid beetles with large differences in size, *Anoplophora chinensis*, *Monochamus saltuarius* and *Saperda populnea*, were selected for the study. Haemolymph was collected by the newly developed net centrifugation method—in which an inner nylon net is used during centrifugation under optimised conditions, and a relatively small wound is generated on the insect—as well as the traditional tearing method and double centrifugation method. Among the three methods evaluated, the net centrifugation method caused the least damage to cerambycid beetles during the whole operation. This method resulted in the most haemolymph from a single beetle, with the lowest turbidity, mostly pure haemocytes in the precipitate, the clearest haemolymph smears by microscopy and the highest quality of RNA extracted from haemocytes. The net centrifugation method has a high collection efficiency, providing important technical support for haemolymph extraction and entomological research.

## 1. Introduction

The insect family Cerambycidae, belonging to the order Coleoptera, includes 35,000 known species [1]. They are the main forest boring insects. Larvae eat the main trunk and roots, and adults bite the skin of the shoots, resulting in the loss of branch tips and weakening of the tree. In severe cases, the whole plant may die. They have caused serious harm to a variety of tree species, such as *Pinus*, *Populus* and *Salix*, in China, affecting forestry development and security [2].

Advances in molecular biology techniques have prompted increasing research on various tissues, organs and extracellular fluids of insects, particularly physiological and biochemical research. Owing to its roles in body fluid circulation, growth and development, many studies have focused on insect haemolymph. For example, Knutelski et al. identified new compounds with antibacterial effects in the haemolymph of *Rhynchophorus ferrugineus* [3]. An analysis of the haemolymph and intestinal flora revealed the cause of lethality of fungal taxa in *Musca domestica* [4]. Secondary metabolites have been identified in *Ulomoides dermestoides* haemolymph [5]. Levi-Zada et al. provided new ideas for benign pest control and management by exploring the relationship between insect behaviours, such as chirping, and pheromones in the haemolymph [6]. In addition, insect haemolymph also plays an important role in predator defence, thermoregulation, gas exchange and so on [7].

Haemolymph research has prompted an increasing focus on collection techniques and methods. Insect haemolymph collection methods have been reviewed [8]. Most collection methods have been developed for larvae or soft-bodied adults. For example, the tearing method has been used for haemolymph collection from *Spodoptera exigua* larvae in analyses of host plant immunity [9], and the wound collection method has been used for haemolymph collection from *Culex quinquefasciatus* adults to analyse the lipophorin sequence and the transcriptional profile [10]. Xu et al. used the double centrifugation method in analyses of changes in serine proteases in *Anopheles dirus* haemocytes after infection with an incompatible rodent malaria parasite [11]. However, in the collection process, adult cerambycid beetles present many difficulties owing to their hard sheathed wings, low haemolymph content, low purity, easy melanisation, etc. [12]. For molecular experiments, it is difficult to obtain a sufficient amount of haemolymph [13]. Few studies have focused on the optimisation of haemolymph collection from adult cerambycid beetles.

To address the lack of effective techniques, we developed an efficient new haemolymph collection method named net centrifugation. Three species with obvious differences in body size, *Anoplophora chinensis*, *Monochamus saltuarius* and *Saperda populnea*, were selected as experimental objects for the optimisation of speed, collection amounts and purity. The newly developed method was compared with methods commonly used for coleopteran haemolymph collection, such as the tearing method and double centrifugation method [14].

## 2. Materials and Methods

### 2.1. Insects

Adults of three cerambycid beetle species with obvious size differences were collected. *A. chinensis* adults were collected from Pingtan, Fujian Province (Figure 1A); *M. saltuarius* adults were collected from Fushun, Liaoning Province (Figure 1B); and *S. populnea* adults were collected from Lhasa, Tibet Province (Figure 1C). *A. chinensis* adults were nearly 2 times larger than *M. saltuarius* adults and 3–4 times larger than *S. populnea* adults although they were all adults of cerambycid beetle species. In total, 48 *A. chinensis* adults, 57 *M. saltuarius* adults and 390 *S. populnea* adults were used for this study. Cerambycid beetles in this study were matched for body size, and all of them were females.

### 2.2. Sample Preparation 

Insects used in molecular experiments are typically cleaned, frozen with liquid nitrogen and kept in a −80 °C refrigerator. All cerambycid beetles used in this research were frozen at −80 °C and thawed before haemolymph collection. For extraction from live cerambycid beetles, specimens were anaesthetized by exposure to cold temperatures (−18 °C) for 120 s and then immediately soaked in 75% alcohol for 30 s; haemolymph was then extracted after wiping with cotton. Except for the extraction process, other processes were the same. As a result, the differences in the measurement of parameters were mainly due to the three methods.

### 2.3. Methods 

#### 2.3.1. Net Centrifugation Method 

A new method for haemolymph extraction was proposed, the net centrifugation method. The following tools were used: 1.5 mL and 15 mL cooled sterile collection tubes (Axygen, Union City, CA, USA), 15 μm sterile nylon net (Shining Industrial Enterprise (China) Co., Ltd., Jiangsu, China), laboratory film (Parafilm, Neenah, WI, USA), 10 cm sterile fine-tipped forceps (Beijing Tricision Biotherapeutics Inc, Beijing, China), 10 cm sterile ophthalmic scissors (Beijing Tricision Biotherapeutics Inc), 75% ethanol (Shandong Annjet Co., Ltd., Shandong, China) and centrifuges (Thermo Fisher, Waltham, MA, USA). 

First, the volume of collection tube A was selected according to the sizes of cerambycid beetles, and a nylon net of the corresponding size was cut. A slightly smaller collection tube B or other rod of suitable size was then used to press the nylon net against the edges of collection tube A, so that the nylon net formed a suitable storage space in collection tube A. Third, the nylon net was fixed on collection tube A with sealing film, and collection tube B was removed. Finally, one pair of antennae and one pair of forelegs were cut off the cerambycid beetles. When a drop of haemolymph spilled out of any of the broken parts of the beetle, the individual was placed head down into the collection tube for centrifugation. A 10 mL collection tube was used for *A. chinensis* (Figure 2A), and 1.5 mL collection tubes were used for *M. saltuarius* and *S. populnea* (Figure 2D,G). A gap between the head of the cerambycid beetle and the bottom of the nylon net prevented damage caused by head impact. According to a gradient experiment, the rotation speed for *A. chinensis* was set to 4000 rpm, the rotation speed for *M. saltuarius* was set to 8000 rpm, and the rotation speed for *S. populnea* was set to 8500 rpm, with a time of 35–40 s and temperature of 4 °C.

#### 2.3.2. Tearing Method 

The tearing method (TM) was used for comparative analyses. This method required a 10–100 μm pipette gun (Eppendorf, Hamburg, Germany), yellow pipette tip (Axygen, Union City, CA, USA), 10 cm sterile fine-tipped forceps, 10 cm sterile ophthalmic scissors, 75% ethanol and 1.5 mL cooled sterile centrifuge tube.

As described previously, fine tweezers were used to tear off the feet and to tear the elytra, or scissors were used to cut the body wall. After generating open wounds, exuded haemolymph was collected directly from the wound with capillary tubes or pipettes [15]. If the amount of fluid collected was small, light finger pressure was applied as needed to increase the volume [16,17]. Alcohol was used to anaesthetise live insects before the experiment and was not used for frozen insects (Figure 2B,E,H).

#### 2.3.3. Double Centrifuge Method

For comparative analyses, the double centrifugation method (DCM) was also used. This method required 1.5 mL and 15 mL centrifuge tubes, 30 G needles (Beijing Tricision Biotherapeutics Inc, Beijing, China), 10 cm sterile fine-tipped forceps, 10 cm sterile ophthalmic scissors, 75% ethanol and centrifuges.

Following the previously described methods, a hole was tied at the bottom of a small centrifuge tube, which was placed in a larger centrifuge tube. Then, an open wound was made (e.g., in the abdomen or dorsal vessel of the insect). When a drop of haemolymph spilled out of any of the broken parts of the beetle, the individual was placed in the small centrifuge tube and centrifuged [18] (Figure 2C,F,I). The speed, time and temperature were consistent with those used for the net centrifugation method.

### 2.4. Measurement of Parameters

#### 2.4.1. Measurement of the Collection Volume

Relatively pure haemolymph fluid was sucked out with a pipetting gun and transferred to a new 1.5 mL centrifuge tube. The amounts of fluid collected under different collection methods were recorded. Haemolymph collected from a single *A. chinensis* and a single *M. saltuarius* were sufficient to measure, while haemolymph from a single *S. populnea* was not. Accordingly, haemolymph samples from ten *S. populnea* were mixed in a tube for quantitation. Each extraction method was subjected to nine repeated experiments for each species. 

#### 2.4.2. Measurement of Precipitate Amounts

EDTA (E1170; Solarbio, Beijing, China) was added to the haemolymph at a haemolymph volume: EDTA volume ratio of 9:1 and centrifuged at 4000 rpm for 20 min at 4 °C. The products of centrifugation were collected [19]. Each species was weighed to the nearest one-millionth of a gram using an electronic balance (Sartorius, Gottingen, Germany). The amount of precipitate collected from a single cerambycid beetle was insufficient to measure for all three species. To ensure a sufficient dosage, haemolymph samples from two *A. chinensis* individuals were mixed in a tube, haemolymph samples from three *M. saltuarius* were mixed in a tube, and haemolymph samples from ten *S. populnea* were mixed in a tube. Samples were then centrifuged and weighed. Each extraction method was subjected to nine repeated experiments for each species.

#### 2.4.3. Measurement of Total Protein 

Total protein was determined using a BCA kit (No. M1806A; Michy, Suzhou, China) following the manufacturer’s instructions. The working solution was prepared, and reagents were added as required. Absorbance was measured at 562 nm and used to calculate the protein content using the formula below. Predetermination was performed before the test, and samples were diluted if the absorbance value was too large. Each sample contained the haemolymph of a cerambycid beetle, with the same amounts used as in Section 2.3.1, and no reagents were added. Each extraction method was subjected to nine repeated experiments for each species.
Cpr (mg/mL) = C1 × (A1 − A2)/(A3 − A2)

Cpr: Content of total protein

C1: Concentration of the standard

A1: Absorbance value of the sample to be tested at 562 nm

A2: Absorbance value of the blank at 562 nm

A3: Absorbance values of standards at 562 nm

#### 2.4.4. Measurement of Turbidity

The quality of the haemolymph differed depending on the collection technique. Therefore, the turbidity of the haemolymph was analysed. The turbidimeter (Qiwei, Zhejiang, China) was calibrated according to the manufacturer’s instructions. Based on the reading range of the instrument, samples were diluted by mixing 5 μL of haemolymph from each cerambycid beetle with 4995 μL of sterile water for a total volume of 5 mL. The measured values were multiplied by the dilution to obtain the final turbidity metric. Each sample contained the haemolymph of a cerambycid beetle, with the same amounts used as in Section 2.3.1, and no reagents were added. Thirty biological replicates were performed for each collection method and each cerambycid beetle species, and three technical replicates were performed on the basis of each biological replicate, for a total of 90 tests for each method and each cerambycid beetle [20].

#### 2.4.5. Observation of Haemolymph Smears 

The collected haemolymph samples were stained with Wright-Giemsa staining solution (No. G1020; Solarbio Life Science, Beijing, China). For staining, 5 μL of haemolymph was dropped onto a slide, and a thin slice was taken to spread the liquid. When dried, 2–3 additional drops of Wright-Giemsa staining solution were applied to cover the entire smear for 1–2 min. An equal volume of phosphate buffer (pH 6.4) (ZCA-BP545; Shanghai Zzbio Co., Ltd., Shanghai, China) was added, the slide was shaken gently, and samples were mixed thoroughly with the staining solution. The cells were stained for 3–5 min, washed with distilled water, blotted dry and observed by microscopy (Leica, Wetzlar, Germany) with a 100× oil objective [21,22,23]. Each sample contained the haemolymph of a cerambycid beetle, with the same amounts used as in Section 2.3.1, and no reagents were added. Three slides were observed for each species and each method. Five visual fields were selected randomly from each slide.

#### 2.4.6. RNA Extraction and Quality Control of Haemocytes

The precipitate was collected following the method described previously (Section 2.3.2), and the precipitate contained the haemocytes. RNAs from haemocytes were extracted using the EASYspin Plus RNA Extraction Kit (No. RN28; Aidlab, Beijing, China). Each method was repeated 12 times for each cerambycid beetle species. The collected RNA was used for standard curve determination using the NanoDrop 8000 (Thermo Fisher). The curve must have a single peak at 260 nm. A concentration of greater than 100 μg/mg is conducive to subsequent molecular experiments. With respect to purity, 260/280 and 260/230 ratios of 1.8–2.2 were considered suitable. After the requirements were met, gel electrophoresis (Thermo Fisher, Waltham, MA, USA) was performed to test the integrity of RNA [24]. 

### 2.5. Statistical Analysis

Statistical analyses were performed using Origin version 2023 (OriginLab, Northampton, MA, USA). Comparisons between each method and each cerambycid beetle species were performed using ANOVA, and normality was evaluated using the Shapiro-Wilk test. All data are presented as means ± SD.

## 3. Results

### 3.1. Comparison of Collection Volumes 

The amounts of haemolymph collected from the three species under different extraction methods are shown in Figure 3A. For all three species, the volume of liquid collected was highest for the net centrifugation method and lowest for the tearing method, with significant differences in volume between methods (*p* < 0.001). In addition, there was a significant difference in volume between the net centrifugation method and the double centrifugation method for *M. saltuarius* and *S. populnea* (both *p* < 0.05). For *A. chinensis*, larger sample volumes were obtained by net centrifugation than by double centrifugation; however, the difference was not significant.

### 3.2. Comparison of Precipitate Amounts

Precipitate amounts obtained from the three species under different extraction methods are shown in Figure 3B. For all three species, the double centrifugation method generated the largest precipitate amount, with visible impurities, followed by the net centrifugation method and the tearing method. The difference in precipitate amounts between the double centrifugation and tearing methods was significant (*p* < 0.05). There was no significant difference in precipitate amounts between the net centrifugation and tearing methods. In addition, the difference between the net centrifugation method and the double centrifugation method was significant (*p* < 0.05) for *M. saltuarius* and *S. populnea*. For *A. chinensis*, the collection volume was lower for the net centrifugation method than for the double centrifugation method; however, the difference was not significant.

### 3.3. Comparison of Total Protein Contents 

The amounts of total protein collected from the three species under different extraction methods are shown in Figure 3C. The total protein content was lowest for the net centrifugation method and highest for the tearing method in *S. populnea*; total protein contents obtained by the net centrifugation and double centrifugation methods differed significantly from the total protein content obtained by the tearing method (*p* < 0.01). For the other two species of cerambycid beetles, there were no significant differences in the total protein content among the three collection methods.

### 3.4. Comparison of Turbidity

The turbidity estimates for the three species under different extraction methods are shown in Figure 3D. In *A. chinensis*, turbidity was highest for the double centrifugation method, followed by the net centrifugation method and tearing method. Turbidity differed significantly between the double centrifugation method and the other two methods (*p* < 0.01), and there was no difference between the net centrifugation method and the tearing method. In the other two species, turbidity was lowest for the net centrifugation method, followed by the double centrifugation method and tearing method. There were significant differences in turbidity among the three methods (*p* < 0.01).

### 3.5. Comparison of Haemolymph Smear Quality

The haemolymph smear results for the three species under different extraction methods are shown in Figure 4. Similar results were obtained for the three cerambycid beetle species. The haemolymph obtained by net centrifugation contained haemocytes with relatively intact morphologies. Visual turbidity was lowest by this method (Figure 4A,D,G). Haemolymph smears collected by the tearing method showed more fat cells in the field of view (Figure 4B,E,H). The haemolymph smear obtained by double centrifugation showed more blood coagulation after staining and greater heterogeneity (Figure 4C,F,H) [25].

### 3.6. RNA Collection and Quality Control Results

The RNA quality estimates for the three species under different extraction methods are shown in Table 1. We counted the qualified number of each measurement and then calculated the qualified number percentage of the total replicates for each method and each cerambycid beetle (Table 1). The success rates of haemolymph collected by the three methods differed among species. For all species, concentration and purity were highest for the net centrifugation method. We selected RNA with a satisfactory calibration curve for gel electrophoresis. These gel electrophoresis results were consistent across all repetitions for each method and cerambycid beetle. Representative agarose gel electrophoresis images are shown in Figure 5, revealing that the tearing method resulted in the highest degree of degradation, followed by double centrifugation method (they showed the smear), when suitable haemolymph was collected from *A. chinensis*. Compared with bands obtained by the net centrifugation method, the gel maps for the tearing method showed additional lighter bands. For *M. saltuarius*, we detected two clear bands on the gels obtained by all three collection methods. For *S. populnea*, the intensity of two bands was highest for the net centrifugation method, followed by the double centrifugation method, and the band obtained by the tearing method was very weak (Figure 5) [26].

## 4. Discussion

Haemolymph participates in whole-body fluid circulation in cerambycid beetles and plays very important roles in growth, development, mating, egg laying and other processes [27]. To address limitations of established techniques, the development of an efficient and simple method for haemolymph collection from adult cerambycid beetles is necessary and can lay a foundation for biological research, as well as pest prevention and control. 

### 4.1. Influence of the Physiological Structure of Adult Cerambycid Beetles on Haemolymph Extraction

Adult cerambycid beetles have a hard shell with elytra completely covering the wings and abdomen. Haemolymph extraction by producing an open wound is limited by the hard shell; the process has the potential to damage the specimen and is laborious [28]. In addition, the haemolymph content is low with easy melanisation. In most nymphs and adults, haemolymph accounts for less than 20% of the body weight, and the cell content is generally less than 2% [29]. The haemolymph extraction time needs to be strictly controlled. A long time period is likely to result in melanisation, making samples unusable in subsequent analyses. Moreover, the haemolymph is in contact with all internal organs and tissues; it transports nutrients, removes metabolic waste and performs immune functions [30]. Therefore, when an internal organ is damaged, the haemolymph will be contaminated. At the same time, there are more fat bodies in the haemocoels of insects, which are relatively large organs distributed in the whole body [31]. They are preferentially distributed under the coat, surrounded by intestinal and reproductive organs, and are the most common components in the internal anatomy [32]. Hence, the mixing of fat with haemolymph can also affect the purity of extracted haemolymph.

### 4.2. Application of Different Methods for the Extraction of Haemolymph from Cerambycid Beetles

The most commonly used methods for haemolymph collection from Coleoptera are the tearing method and double centrifugation method. When using the tearing method to collect cerambycid beetle haemolymph from the dorsal vessels, the elytra need to be cut off first, which damages the whole body of the beetle [33]. Large cerambycid beetle shells are extremely hard, making haemolymph collection from the thorax or abdomen difficult. Only a small amount of haemolymph fluid flows out, even with proper compression, making the process time-consuming and labour-intensive, with easy melanisation.

When using the double centrifugation method to collect haemolymph, most studies apply centrifugation after making open wounds on the dorsal blood vessels or abdomen of insects. However, the back and abdomen are close to the stomach, malleus and other organs that store food and excrement in the midgut and hindgut, and high-speed centrifugation in the presence of wounds in the surrounding body wall will cause repeated collisions with the hard wall of the centrifuge tube, leading to internal and external damage and increased contaminants [34,35,36]. Proposals to reduce contamination include reducing the centrifugation speed, prolonging the centrifugation time and adding chemical reagents to the bottom of the tube to prevent melanisation [37]. However, the addition of reagents would affect the first round of haemolymph collection and may result in reactions with reagents needed in subsequent experiments; accordingly, no reagents were added in this experiment. In addition, in the current study, the lack of a standard range of centrifuge speeds for taxa of different sizes made it difficult to collect longhorn beetle haemolymph by the double centrifugation method.

Therefore, on the basis of the double centrifugation method, we developed a new haemolymph collection method, the net centrifugation method. First, in the collection device, we replaced the inner centrifuge tube used in double centrifugation with a nylon net (15 μm) to ensure that large material is filtered out, and that blood lymph and haemocytes pass through. Moreover, the nylon net has a smooth surface and weak water absorption ability, avoiding the accumulation of lymph residue during collection. Second, we adjusted the open wound site in cerambycid beetles. We removed a pair of antennae and a pair of forefeet, and then performed centrifugation. This produces a relatively small wound without substantial harm to the insect. Third, haemolymph collected at a low rotation speed was relatively pure. However, the amount of haemolymph collected in a short period of time was very small. Increasing the rotation speed slightly can ensure the purity of the haemolymph collected at a sufficient volume. Therefore, the optimum centrifugal speeds for the three species of cerambycid beetles were obtained by gradient experiments, indicating that speeds of 4000 rpm for larger cerambycid beetles, such as *A. chinensis*, and approximately 8500 rpm for smaller cerambycid beetles, such as *S. populnea*, are effective. We further identified a range of centrifugation speeds for cerambycid beetles with similar sizes; the critical centrifugation conditions for haemolymph extraction were 4500–8500 rpm, 35–40 s and 4 °C for cerambycid beetles between the size of *A. chinensis* and *S. populnea* (appropriate adjustment can be made according to the size of the cerambycid beetles). These conditions can reduce contamination caused by the breakage of insects while ensuring a sufficient collection volume, providing a convenient method applicable to various species. Finally, when comparing the three collection methods, we assessed the quality of RNA extracted from haemolymph cells, providing an index for the assessment of haemolymph quality and for the selection of the most applicable extraction methods [26]. This parameter was not reported in the previous haemolymph quality assessment.

### 4.3. Improvement in Haemolymph Extraction from Cerambycid Beetles by the Net Centrifugation Method

Based on several indicators, the best overall results were obtained by the net centrifugation method, which yielded high-quality samples. In terms of collection amounts and time, the net centrifugation method showed obvious advantages over the other two methods, and no haemolymph melanisation was detected during the collection process. Compared with the double centrifugation method with the same speed but slightly lower collection volumes, the haemolymph collected by the net centrifugation method showed a significantly higher purity, without excessive impurities in the precipitate after centrifugation. We believed that if the coloured impurity was removed, the net centrifugation method would generate more precipitate than the double centrifugation method. Further, the net centrifugation method had clearer haemolymph smears [38,39,40]. Blood coagulation was observed during the microscopic examination of haemolymph collected by the double centrifugation method, and the field of view was unclear, indicating that the cerambycid beetles were stimulated and injured [41,42,43]. In addition, the quality of RNA extracted from haemocytes by net centrifugation was better than that of RNA extracted by the other two methods, indicating that the method is suitable for molecular experiments involving haemolymph. 

However, this study had some issues that should be addressed in the future. For example, the observed total protein contents were not consistent with our expectations. The total protein content should be positively correlated with the cell content in the haemolymph [32]. Haemolymph with higher turbidity contains more substances and more protein [44]. For *A. chinensis* and *M. saltuarius*, we did not obtain a significant difference in total protein content despite the difference in turbidity among collection methods. This may be explained by the sample size. It is also possible that during haemolymph collection, fat bodies were mixed within the fluid. The acidity of the fat bodies altered the required alkaline conditions for protein content assays using the BCA method (i.e., the protein could not reduce Cu^2+^ to Cu^+^, and the colour reaction was weak) [45,46]. In addition, there is no guarantee that the collected haemolymph is free of contamination. In general, among the three methods evaluated, overall performance of the net centrifugation method was the best. First, it caused the least damage to cerambycid beetles during the whole operation. Second, this method resulted in the most haemolymph from a single beetle, with the lowest turbidity and mostly pure haemocytes in the precipitate. Third, it produced the clearest haemolymph smears by microscopy. Fourth, the highest quality of RNA was extracted from haemocytes through the net centrifugation method. Further modification of the equipment is needed for the collection of large amounts of pure and nonmelanised haemolymph. Expertise in a variety of disciplines is needed to identify materials able to completely filter out fat and other impurities, while ensuring the smooth passage of haemolymph.

### 4.4. Versatility of the Net Centrifugation Method in the Extraction of Haemolymph from Cerambycidae

The majority of methods for haemolymph collection are aimed at a specific species, which limits the scope of application [47,48,49]. In Cerambycidae, the smallest species in Europe, *Gracilia minuta*, is 4–6.5 mm long and 1–2 mm wide, and the largest species in the family worldwide, *Titan beetle*, is 16.7 cm long. Other than size, species in the family do not show substantial structural differences. Therefore, in this study, three cerambycid beetles with obvious differences in size were selected to explore whether the net centrifugation method was applicable to different species of cerambycids. A comparative analysis revealed that the net centrifugation method had the best overall collection performance in the three species, demonstrating the versatility of the method.

## 5. Conclusions

The collection of cerambycid beetle haemolymph has always been challenging. The net centrifugation method utilised modified materials and procedures to improve the collection volume and purity of adult cerambycid beetle haemolymph to a certain extent. The method enabled rapid haemolymph collection with inexpensive and widely available materials. The newly developed method is an innovation in the field of haemolymph collection, providing a convenient approach for research and the development of pest control strategies. 

## Figures and Tables

**Figure 1 insects-14-00029-f001:**
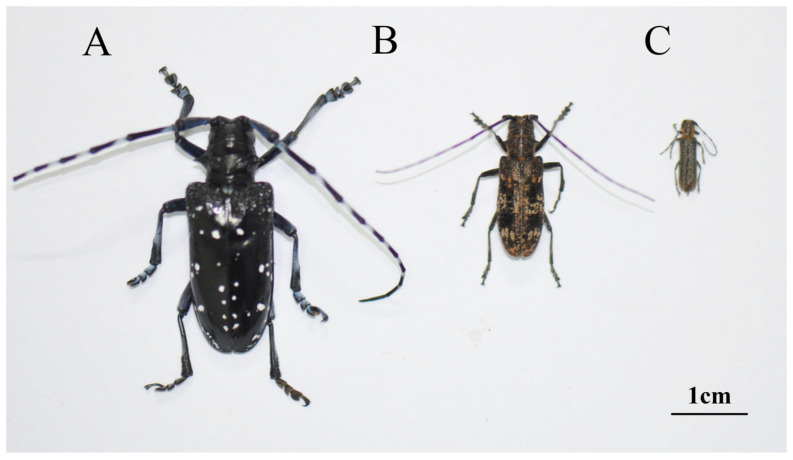
Images of three cerambycid beetle species. (**A**): *Anoplophora chinensis*, (**B**): *Monochamus saltuarius*, (**C**): *Saperda populnea*.

**Figure 2 insects-14-00029-f002:**
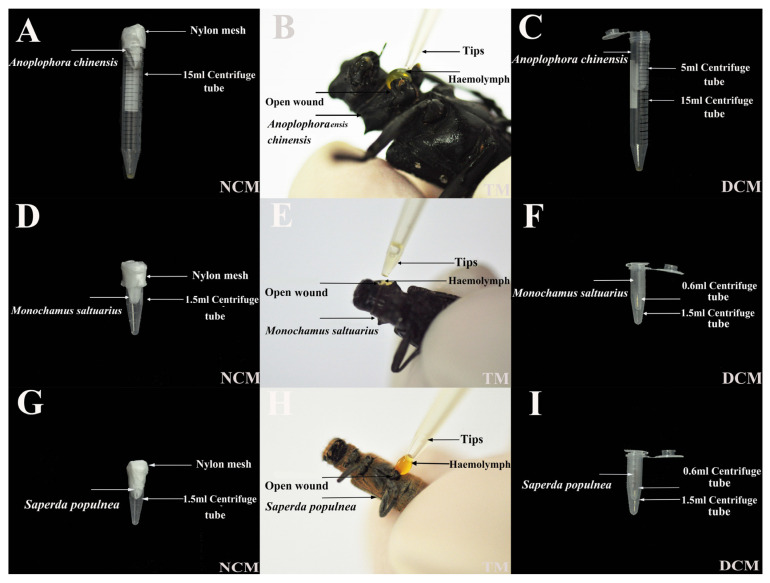
Overview of three methods used to extract haemolymph from adult cerambycid beetles. (**A**,**D**,**G**): net centrifugation method (NCM); (**B**,**E**,**H**): tearing method (TM); (**C**,**F**,**I**): double centrifugation method (DCM).

**Figure 3 insects-14-00029-f003:**
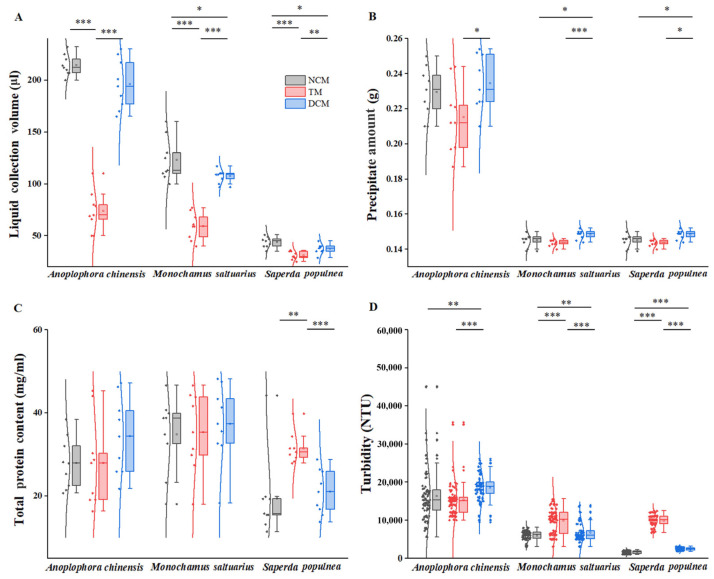
Results of three haemolymph collection methods for three species. (**A**): collection volumes; (**B**): precipitate amounts; (**C**): total protein contents; (**D**): turbidity. NCM, net centrifugation method; TM, tearing method; DCM, double centrifugation method. Samples, black dots; ±SD, hollow dots; median, middle line inside each box; IQR (interquartile range), the box containing 50% of the data; whiskers, 1.5 times the IQR. Normal of distribution, curved line. * *p*  <  0.05; ** *p*  <  0.01; *** *p*  <  0.001.

**Figure 4 insects-14-00029-f004:**
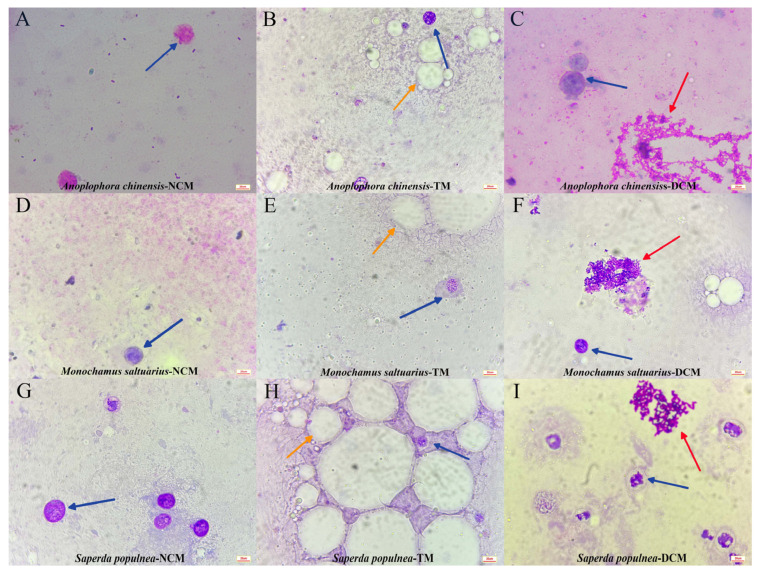
Microscopic examination of smears from three cerambycid beetle species for different collection methods under a 100× oil immersion lens. Blue arrows point to haemocytes; yellow arrows point to fat cells; red arrows point to blood coagulation. (**A**–**C**): Microscopic examination of smears from *Anoplophora chinensis* for NCM, TM and DCM under a 100× oil immersion lens. (**D**–**F**): Microscopic examination of smears from *Monochamus saltuarius* for NCM, TM and DCM under a 100× oil immersion lens. (**G**–**I**): Microscopic examination of smears from *Saperda populnea* for NCM, TM and DCM under a 100× oil immersion lens.

**Figure 5 insects-14-00029-f005:**
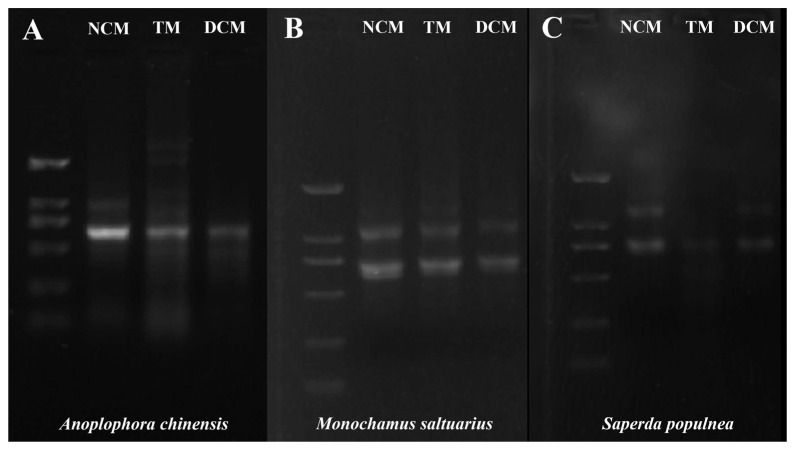
Gel electrophoresis images for haemolymph RNA obtained by three collection methods in adults of three cerambycid beetle species. (**A**): Gel electrophoresis images for haemolymph RNA obtained by three collection methods of *Anoplophora chinensis. (***B**): Gel electrophoresis images for haemolymph RNA obtained by three collection methods of *Monochamus saltuarius*. (**C**): Gel electrophoresis images for haemolymph RNA obtained by three collection methods of *Saperda populnea.*

**Table 1 insects-14-00029-t001:** Quality indexes of dissolution profiles for three species of cerambycid beetles under different collection methods (N = 12).

Species	Method	Qualified Curve		Qualified Concentration		Qualified 260/280		Qualified 260/230	
		n	%	n	%	n	%	n	%
*Anoplophora chinensis*	NCM	12	100.0	12	100.0	12	100.0	10	83.3
	TM	12	100.0	12	100.0	11	91.7	10	83.3
	DCM	9	75.0	9	75.0	10	83.3	9	75.0
*Monochamus saltuarius*	NCM	12	100.0	12	100.0	12	100.0	9	75.0
	TM	10	83.3	10	83.3	8	66.7	9	75.0
	DCM	11	91.7	12	100.0	10	83.3	8	66.7
*Saperda populnea*	NCM	11	91.7	12	100.0	9	75.0	8	66.7
	TM	5	41.7	6	50.0	5	41.7	5	41.7
	DCM	8	66.7	11	91.7	7	58.3	5	41.7

Qualified curve: a single peak at 260 nm; qualified concentration: ≥100 μg/mg; qualified 260/280: 1.8–2.2; qualified 260/230: 1.8–2.2; the N stands for the replicates for each method and each cerambycid beetle; the n stands for the number of qualified for each method and each cerambycid beetle; the % stands for the qualified percentage.

## Data Availability

The data presented in this study are available in the insert article.

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
