# Peer review of "An Efficient and Simple Method for Collecting Haemolymph of Cerambycidae (Insecta: Coleoptera) Adults"

_insects, 2022, doi:10.3390/insects14010029_

Round 1

Reviewer 1 Report

This is a very interesting manuscript, however, it needs to be extensively reviewed for English editing, as it is rather difficult to read and follow in its current state.

Please rephrase the title: An efficient and simple method to collect haemolymph in adult cerambycid beetles

From section 2.1 it is unclear if only 3 insects were used in this study (one from each species), and whether 3 collections techniques were tried on each single insect. If the 3 techniques were performed on each, how can the authors be sure that the performance of the second and third technique were not influenced by the preceding techniques performed on the same insect? The authors should specify in section 2.1 how many insects of each species were used in total.

In section 2.2, please specify what was used as a “sealing membrane”. Please be more specific with the material that you used. For example, please specify the model of scissors and tweezers that were used. What does the 800 stand for in the nylon mesh? 800 microns? Perhaps the authors could provide the exact material that was purchased so that the method is more easily reproduced by other researchers (adding the item number, for example). Please make sure that you use the right company name as well (do you mean Thermo Fisher?).

The authors mention using alcohol to clean and anesthetise the live insects. Please describe which alcohol was used and how this was performed (Dipping in alcohol? For how long? Etc.).

The authors mention using also -80 C frozen beetles. Please provide a comparison between live and frozen beetles in terms of method performance, etc. Also, what are the additional steps to perform with frozen beetles? Should they be thawed first? If so, for how long? Etc.

The methods section needs to be rearranged, as information is misplaced and extremely difficult to follow. For example, at the end of 2.3.1 Measurement of the collection volume, there is a sentence mentioning the number of repeated experiments as well as the number of S. populnea samples pooled together. In section 2.3.2, the number of pooled A. chinensis and M. saltuarius is mentioned. Why are these informations in two different sections? All of this should be grouped in the same section, to specify how many insects of each species were used, how many in each experiment, how many times each experiment was performed, how many with live vs frozen, etc. Please review all of the methods section and make sure that the appropriate information is found in each section and not separated in different sections.

At 2.3.2, the authors mention mixing the samples with EDTA. Was this done for all samples used in the subsequent sections? This is very confusing, as in 2.3.6, the authors mention using a specific number of insects for RNA extraction, hence it seems unlikely that EDTA was added to those. Please make sure that the sample preparation for each technique is very clear.

At 2.3.5, please specify if tap water was used for rinsing, or if it was another type of water (demineralized, or other). Please also specify how many slides were observed for each species, if insects were pooled to prepare slides, if EDTA was added to those samples, etc.

There is no section on statistical analysis. One must be added, with details on the tests used, pre-tests for normality of distribution (if required), data representation (Is SEM or SD used in figures and text?).

Section 3 Results start with two sentences that need to be removed (This section may be divided by subheadings, etc.).

In figure 3, what does the curved line beside each boxplot represent? Also specify if SEM or SD were used in the boxplots.

In figure 4, please add in the legends what the different arrows are pointing at.

L288 is a good example of the need of careful review for English editing: this sentence suggests that the hemolymph contains only 2% of the blood cells, and hence that the other 98% should be elsewhere. However, what the authors likely mean is that hemocytes compose 2% of the total hemolymph volume (hence, the “hematocrit” is very low in insects compared to other animal species).

I am not sure that “lymphocytes” is the correct name for hemolymph cells in beetles. Perhaps hemocytes would be more appropriate. If so, please change throughout.

Reviewer 2 Report

The authors report a novel method for collecting adults of cerambycid beetles haemolymph.

I am not sure how efficient and simple it is as it requires a centrifuge and tubing and more utensils than a simple cut and extract method (tearing method). They both euthanize the specimen. 

I have difficulty to follow the method with the black image. Using black as a background for an image makes it very difficult to see the content. I would suggest using white background.

I also don't understand some of the results.

Please see comments below. Thank you

The title should inform what the target organism is

“An efficient and simple method for collecting adults of cerambycid beetles (Coleoptera: Cerambycidae) haemolymph

 “At present, there is few efficient method to”  -  correct to “At present, there are few efficient methods to …”

 I think the  existing methods are efficient. The advantage of yours seems to be quality, not volume.  The text should emphasize this.

“Cerambycidae, belonging to Coleoptera, includes 35,000 known species” – correct to “The insect family Cerambycidae, belonging to the order Coleoptera, includes 35,000 known species”

 There was no significant difference in precipitate amounts between the net centrifugation and tearing methods, so quantity is not an advantage to your method. It seems quality is and therefore this should be highlighted in the text.

Figure 5. Gel electrophoresis images for haemolymph RNA obtained by three collection methods in three species of adults of cerambycid beetles.

The gel shows one sample for each organism. The authors did replicate the study ( nine repeated experiments). We should know if this result is consistent across samples tested. One single individual is insufficient to extrapolate results. This is a problem in this paper.

Table 1 and Figure five are the most important results of the performance of this method, i.e., quality of RNA. However only one specimens out of all the replicates done is shown. We don't know if this is consistent or just happened once. 

Table 1.  We need to know if these are significant or not. There is no test shown. And how was this measured? You need to state this in the legend. I understand you have it in the methods but the figures need to be self-explanatory whithout having to go back to the text 

I am not sure what he percentages mean. 260/280 absorbance  values (at least in a Nadodrop) reflect protein contamination. 260/230 absorbance reflect contamination with other compounds. So I am not sure what these %s mean because we don’t know how these were calculated.

you mention in the methods "With  respect to purity, 260/280 and 260/230 ratios of 1.8–2.2 were considered suitable. After the requirements were met, the gel electrophoresis experiments were prepared to test the integrity of RNA"

Where are the 1.8-2.2 values? those are the values that tell the purity of the samples. I don't know what the percentages you show are.

Reviewer 3 Report

This is a fine manuscript that improved methodology of hemolymph extraction. While i don't have any major comments on experiments, my concern was regarding the similarity between the methods while extracting compounds of interest (Proteins, Hemocytes) However, they do address the shortcomings of the study in the final paragraph of the discussion. I think an additional sentence (or modify what is written now) to reflect that while each method has its own merits, our method in comparison is better at 1,2,3.. would make it clearer. I have some minor comments written below.

line 50- italicize

line 58: talks about mosquitoes and bees, and as an example in the next sentence talks about a lepidopteran.

line 72: giving common names would be good for readers

line 83: would be a good thing to explain the differences ins ample size due to size differences of the beetles

were they all males or females or a mix?

line 140: really hard to read the text on the figures, also the black background makes it harder to read.

256: again, i cant read the legends on the figures.

Round 2

Reviewer 1 Report

I wish to congratulate the authors for the diligent correction of their manuscript. 

L261: I believe that the authors mean that the curved line represents the normal distribution (not normality).

Reviewer 2 Report

Thank you for addressing the questions.

My question regarding if the results are consistent across replicates and organisms tested remains. 

Authors should state if this is the case as they only show one gel, one specimens, one species, one result.

Point 8: Figure 5. Gel electrophoresis images for haemolymph RNA obtained by three collection methods in three species of adults of cerambycid beetles.

The gel shows one sample for each organism. The authors did replicate the study ( nine repeated experiments). We should know if this result is consistent across samples tested. One single individual is insufficient to extrapolate results. This is a problem in this paper.

Response 8: Thanks for your constructive comments. In most cases, after we illustrated our results of gel electrophoresis, we only use a representative picture to show. As a further demonstration, we provide some references:

(1) HUANGWen-Yu; LUYing; YIN Can-Can; MA Gao-Feng; LIU Xiao; SHAO Huang-Fang ;LI Meng-Nan;SUN En-Tao. A suitable method for isolating total RNA and preserving samples from mosquitoes. Chinese Journal of Applied Entomology 2019,56(06),1430-1436.

(2) Anastasia M.W. Cooper; Zhitao Yu; Marie Biondi; Huifang Song; Kristopher Silver; Jianzhen Zhang; Kun Yan Zhu. Stability of double-stranded RNA in gut contents and hemolymph of Ostrinia nubilalis larvae. Pesticide Biochemistry and Physiology 2020, 169, 104672.

The title should say what the organisms are (you are assuming we know these are insects). Readers might need that information. 

Response 10: Thanks for your constructive comments. We have made some changes to the Table 1 according to your requirements. We could obtain the results by measuring through NanoDrop 8000 and feedback was presented in the form of a picture (as shown in the following figure) , no other tests need to be done. The results were summarized and analyzed and then showed in Table 1.​

Yes, I am familiar with the Nanodrop image you showed, however those values are not the ones you list in your table. You list percentages. My question is: how were these obtained? You list "These %s mean the four indicators’ qualification rate for each method and each cerambycid beetles species." and also "After obtainted the results of 260/280 and 260/230, we then calculated number that meets the requirement directly. So we didn’t present the 1.8-2.2 values, just show the qualification rate for each method and each cerambycid beetles species.

I am sorry. I just don't understand this information, and how was it obtain. You should list "qualified 260/280 measurements where converted to percentages by...."
